# Violence and hepatitis C transmission in prison—A modified social ecological model

**Hossain M. S. Sazzad**[1]*, **Luke McCredie**[2], **Carla Treloar**[3], **Andrew R. Lloyd**[1], **Lise Lafferty**[1,3]

**1** The Kirby Institute, UNSW Sydney, Sydney, Australia, **2** Centre for Health Research in Criminal Justice, Justice Health, Sydney, New South Wales, Australia, **3** Centre for Social Research in Health, UNSW Sydney, Sydney, Australia

* hsazzad@kirby.unsw.edu.au

## Abstract

### Background

Transmission of hepatitis C virus (HCV) among the prisoner population is most frequently associated with sharing of non-sterile injecting equipment. Other blood-to-blood contacts such as tattooing and physical violence are also common in the prison environment, and have been associated with HCV transmission. The context of such non-injecting risk behaviours, particularly violence, is poorly studied. The modified social-ecological model (MSEM) was used to examine HCV transmission risk and violence in the prison setting considering individual, network, community and policy factors.

### Methods

The Australian Hepatitis C Incidence and Transmission Study in prisons (HITS-p) cohort enrolled HCV uninfected prisoners with injecting and non-injecting risk behaviours, who were followed up for HCV infection from 2004–2014. Qualitative interviews were conducted within 23 participants; of whom 13 had become HCV infected. Deductive analysis was undertaken to identify violence as risk within prisons among individual, network, community, and public policy levels.

### Results

The risk context for violence and HCV exposure varied across the MSEM. At the individual level, participants were concerned about blood contact during fights, given limited scope to use gloves to prevent blood contamination. At the network level, drug debt and informing on others to correctional authorities, were risk factors for violence and potential HCV transmission. At the community level, racial influence, social groupings, and socially maligned crimes like sexual assault of children were identified as possible triggers for violence. At the policy level, rules and regulations by prison authority influenced the concerns and occurrence of violence and potential HCV transmission.

**Data Availability Statement:** As is standard practice with publications reporting on qualitative data, the data will not be made available. The restriction is imposed by ethics review committee. The address of the committee is as follow: Human

Ethics Team| Human Research Ethics Team
Research Ethics and Compliance Support (RECS) |
UNSW SYDNEY| Level 3, Rupert Myers Building
Malory Plummer| Human Ethics Officer| T+61 2
9385 7333.

**Funding:** This project was funded by the National
Health and Medical Research Council
(APP1016351) and NSW Health. The funders had
no direct role in the conduct of this research or the
presentation of findings. ARL is supported by
NHMRC Fellowship (APP1137587). The Kirby
Institute and the Centre for Social Research in
Health are funded by the Australian Government
Department of Health and Ageing. The funders had
no role in study design, data collection and
analysis, decision to publish, or preparation of the
manuscript.

**Competing interests:** The authors have declared
that no competing interests exist.

## Conclusion

Contextual concerns regarding violence and HCV transmission were evident at each level of the MSEM. Further evidence-based interventions targeted across the MSEM may reduce prison violence, provide opportunities for HCV prevention when violence occurs and subsequent HCV exposure.

## Introduction

Hepatitis C virus (HCV) infection is a major public health threat with estimated global prevalence of 1% chronic infection [1]. HCV is a blood-borne virus (BBV), frequently transmitted through unsafe injection practices such as sharing of contaminated equipment, especially in high-income countries [2,3]. Among people who inject drugs (PWID), estimated 52% have detectable antibodies against HCV, and 58% have a history of imprisonment across the world [4]. Multiple factors contribute to the higher prevalence of HCV infection in prison than those in the community with criminalisation of drug use being the major contributor [5]. Globally, HCV antibody prevalence among the prisoner population is estimated to be 15% [6], with surveillance of the Australian prisoner population revealing a 22% prevalence [7]. Injecting drug use within the prison carries a high per injecting episode risk of HCV exposure [8]. This is largely attributed to the lack of access to sterile injecting equipment which leads to frequent sharing of injecting equipment [5,9,10].

Beside injecting risk exposures, several non-injecting risk behaviours including physical violence in which blood-to-blood contact occurs [11], tattooing [12], reuse of barber's shears [13], sexual transmission among males via anal sex [14], and in rare occasions vaginal intercourse [15] have been linked to transmission of HCV within the prison setting. However, the contextual concerns around transmission of HCV associated with violence has been poorly studied [16,17]. Exposure to bleeding caused by intimate partner violence were independently associated with transmission of HCV [18]. The evidence of HCV transmission following bloody fist fight has been reported [19]. A qualitative study in Australian prisons pertaining to economics of drug use and blood borne virus transmission examined physical violence, drug debt and potential HCV transmission as a complex interrelated issue [20]. Holistic understanding beyond individual risk factors for HCV transmission creates an opportunity to craft appropriate organisation-wide preventive strategies.

The modified social-ecological model (MSEM) is a comprehensive approach to identifying contextual concerns regarding disease transmission considering individual, network, community, public policy levels, and the stage of epidemic [21,22]. The individual level of the model includes biological or behavioural characteristics associated with the vulnerability to acquire or transmit the pathogen and developing infection [23,24]. At the network level, social networks are considered which include interpersonal relationships such as family, friends, neighbours and others that directly influence health and health behaviours that might predispose the transmission [24]. Community level of the framework considers the cultural, economic, religious, geographic lines, prison walls, community norms, stigma, race, code of conduct in prison, or any combination that may bind communities [25]. Policy level examines policies and laws from the stakeholders' perspectives and subsequent decision on programmes to prevent disease transmission [26–28]. Ultimately, the epidemic level is determined by disease burden across different settings [26,29]. The framework has been utilised to characterise BBV infections, including HIV infection, with the aim of guiding development of HIV prevention

strategies [21]. Centres for Disease Control and Prevention, United States (US CDC) has developed a technical package for violence prevention based on the social ecological framework [30]. The MSEM was previously adopted to identify the concerns among vulnerable population groups (i.e., injecting drug users and men who have sex with men) for HIV prevention [21]. Identification of concerns might help to implement further research on specific levels for increased understanding and intervention responses. The MSEM framework enabled useful insights into the complex public health problem of HCV transmission in prison. The objective of this study was to describe the context and concerns among prisoners regarding HCV transmission in prison associated with violence using the MSEM framework.

## Methods

This qualitative study was conducted as part of a broader prospective cohort study, the Hepatitis C Incidence and Transmission Study in prisons (HITS-p) [31,32]. The objective of HITS-p was to estimate HCV incidence and identify risk factors for transmission in the prison setting. Participant enrolment commenced in 2005 across 30 correctional centres in New South Wales (NSW), Australia and concluded in 2014. A total of 590 persons in prison were enroled in this cohort study [9]. Participants enrolled in the HITS-p study were eligible for the qualitative sub-study. The objective of the qualitative sub-study was to understand the broader contexts and concerns regarding HCV transmission in prison. Thirty participants were recruited in the sub-study. Among them, a subset of participant interviews describing contexts and concerns regarding violence in prison and HCV transmission was analysed for this study. The remaining participant interviews focused on decisions about hepatitis C treatment.

Corrective Services NSW, Justice Health and Forensic Mental Health Network, and University of New South Wales human research ethics committees provided approval for the HITS-p cohort study, including the qualitative sub-study.

Prisoners aged 18 years or above who reported either a history of ever injecting drug use or had non-injecting risk behaviours (including tattooing, piercing or fighting), *and* had a documented negative anti-HCV test result in the 12 months prior to enrolment were eligible to participate in HITS-p cohort study. Prisoners with detectable antibodies against HCV, insufficient English or current psychiatric disorder to preclude consent, or were pregnant, were excluded from the study. The qualitative data explored the complex and inter-related nature of practices and environments surrounding HCV risk and potential prevention strategies among prisoners. An interview-based method was chosen to allow participants to fully discuss and explore the context of violence and potential blood-borne viruses including HCV transmission by physical violence in prison.

During HITS-p cohort study, all participants were screened for HCV antibodies and viraemia; and then monitored every 3 to 6 months via blood testing. An interviewer-administered questionnaire was completed at each visit to record both injecting (e.g., frequency of injecting, frequency of sharing/use of personal equipment) and non-injecting risk behaviours (e.g., tattooing, piercing and physical violence).

During 2013–2014 HITS-p study participants were invited to be enrolled in this qualitative study. The qualitative study methods and results adhere to the Consolidated Criteria for Reporting Qualitative Research (COREQ) [33]; see Supporting information S1 COREQ checklist. The psychology-trained HITS-p research nurse (LM), who was engaged in the HITS-p study for several years during which he collected blood samples and conducted behavioural surveys with HITS-p participants, also informed the prisoners about the qualitative study and offered the opportunity to participate. The interviewer (LM) was trained and supervised by the experienced qualitative social researcher (CT) of the HITS-p study. All

participants were selected purposively to represent injecting and non-injecting drug use, and risk exposures among prisoners with and without HCV infection. The nurse explained the purpose of the study and the prisoner's rights to accept or decline the offer (a decision to not participant in the qualitative component had no bearing on their involvement in the larger HITS-p study or their relationships with Corrective Services NSW or Justice Health & Forensic Mental Health Network). When the in-depth interviews were scheduled and the prisoners attended, the nurse reiterated the ethical principles of informed consent and confidentiality, withdrawal without penalty, and the importance of avoiding discussion of specific serious-incidents which would require legislated mandatory reporting to authorities. Written informed consent was obtained. To protect participants' privacy, interviews were conducted in a private clinic room in the absence of correctional officers. Participants received AU$10 into their inmate account for their participation in the interview through the approvedprison inmate banking system to compensate for their time and effort in completing the research interview. This amount was recommended by the research ethics committee as being sufficient to constitute 'reimbursement for time and convenience' (as was stated on the consent form), but insufficient to provide a strong or coercive incentive to participate. In practice, these monies are typically expended for 'buy-ups'–that is purchase of food or toiletry items not otherwise available in the prison.

An interview guide was developed by the authors who are experienced in prison-based health research. Probes were used to facilitate discussions. The interview schedule included topics such as HCV risk perceptions (what risks are perceived by inmates; what risks can be compromised or negotiated and what cannot); participant's knowledge and perception of susceptibility of HCV infection, as a highly prevalent BBV, especially among PWIDs in the prison setting; and injecting and non-injecting risk behaviours, including tattooing and violence. During specific discussion regarding violence, the type of violence, and situations that lead to violence, as well as prisoners' concerns around violence were explored (S1 Appendix). Importantly, how HCV transmission risk was configured in relation to this violence in prison was also discussed. Demographic information was collected from all participants.

The duration of the interviews ranged from 30 to 70 minutes. At the conclusion of each interview, participants were provided with written information about HCV, an opportunity to discuss any further issues with the research nurse, and information about access to the Prison Hepatitis Infoline (a toll free service connecting people in custody with the state's community-based hepatitis organisation).

Interview transcripts have been assessed for data saturation, with no new themes emerging in the final interviews. The responses regarding violence in prison from this subset of participants achieved saturation, hence the interview focus conducted among remaining prison participants shifted to HCV treatment.

Interviews were audio-taped and transcribed verbatim. Transcripts were checked for accuracy against recordings, de-identified and cleaned. The data was then read closely, and a number of themes were identified as relevant to the research questions, specifically relating to violence in prison. The research team then collaborated on the construction of a "coding frame"—a set of organising, interpretive themes to aid analysis [34]. CT and LM developed the first-round coding framework. The coding frame was then used to organise interview data within NVivo 12 (QSR International Pty Ltd. Version 12, 2018). Memos were written between close reading of the transcripts and development of the MSEM coding framework. HS and LL developed the secondary coding framework to elicit responses pertaining to the MSEM. LL conducted secondary coding of random samples to ensure consistency and to establish intrarater reliability. The primary or initial coding frame and the MSEM coding frame are separate projects with distinct analyses to interpret different aspects of HCV risks within the prison

setting. However, the importance of violence was a primary interest of the project and is examined here in a secondary analysis using the MSEM.

The analysis was informed by both a deductive and inductive approach to cover the contexts and concerns regarding HCV transmission following violence considering the MSEM [35]. Specifically, each participant's response was reviewed to examine the specific circumstances of violence in prison, and their concerns regarding HCV transmission through violence at every stage of the MSEM. Consideration of the stage of the epidemic in the social ecological framework highlights the fact that the high burden of HCV infection in the prison setting, impacts on the consequences of prisoner violence (with a heightened risk of transmission) and the importance of recognising this context in prevention strategies. Each aspect of the thematic analysis, that is, the interpretations and meanings drawn from the interview data was critically examined and summarised along with supporting quotes. Quotes are presented by participant number, gender, and nature of risk behaviours—injecting and/or other risk behaviours, including tattooing, piercing and physical violence.

## Results

Twenty-three people in prison participated in this study, eight (35%) of whom were female. During June 2014, 2591 (7.7%) prisoners were female in Australian prisons [36]. The median age of participants was 27 years; range 22–51 years. Thirteen (57%) participants were white and 9 (39%) participants identified as Indigenous. One participant's racial background was not provided. Of this group, 10 had no detectable HCV antibody (not exposed to HCV) at the time of interview, 5 had chronic HCV infection (persistent infection for greater than six months) and 8 had recent HCV infection (Table 1). All acute and chronic HCV infections had been acquired during incarceration. Among 30 participants of the broader qualitative study, the final seven participant interviews focused on decisions about HCV treatment and did not include discussion of violence as a risk factor for HCV transmission. Information is not available in relation to participants refusing invitation to participate.

**Table 1. Demographic and clinical characteristics of the participants (n = 23).**

| Characteristics | Male (n = 15) % | Female (n = 8) % | Total (n = 23) % |
|---|---|---|---|
| **Median age in years (range)** | 27 (22–51) | 30 (23–34) | 27 (22–51) |
| **Education** | | | |
| Primary (grade 1–6) | 1 (6.7) | 1 (12.5) | 2 (8.7) |
| Secondary (grade 7–10) | 7 (46.7) | 6 (75) | 13 (56.5) |
| Higher secondary (grade 11–12) | 3 (20) | 0 (0) | 3 (13) |
| Tertiary education (>grade 12) | 1 (6.7) | 1 (12.5) | 2 (8.7) |
| Technical and further education (TAFE) | 3 (20) | 0 (0) | 3 (13) |
| **Ethnicity** | | | |
| Indigenous | 8 (53.3) | 1 (12.5) | 9 (39.1) |
| Non-Indigenous | 7 (46.7) | 6 (75) | 13 (56.5) |
| **Risk exposures** | | | |
| Only injecting drug use | 6 (40) | 3 (37.5) | 9 (39.1) |
| Other exposures than injection (e.g. tattoo, body piercing, physical violence) | 3 (20) | 1 (12.5) | 4 (17.4) |
| Both injecting and other exposures | 6 (40) | 4 (50) | 10 (43.5) |
| **Hepatitis C clinical status** | | | |
| No HCV antibodies | 5 (33.3) | 5 (62.5) | 10 (43.5) |
| Acute hepatitis C | 6 (40) | 2 (25) | 8 (34.8) |
| Chronic hepatitis C | 4 (26.7) | 1 (12.5) | 5 (21.7) |

Risk factors and behaviours varied among the participants. Ten participants reported only injecting drug use; nine participants reported both injecting drug use and other possible exposure to blood (via tattooing, piercing, violence or haircuts) and four participants reported only other non-injecting related possible exposure to blood (via tattooing, piercing, violence, or haircuts).

Physical violence among participants was a clear concern for potential transmission of blood-borne viruses, including HCV. Importantly, the risks and concerns were varied across different levels of the social ecological model–individual, network, community, public policy and stage of epidemic. The framework provided understanding of risk contexts, and of concerns around violence and HCV and other blood-borne pathogen transmission following violent episodes.

## 1. Individual-level contexts

At the individual level of the socio-ecological framework, the concerns regarding HCV transmission expressed were related to the nature of violence encountered by the prisoners. Participants reported different types of violence in prison. The minimum level of violence was verbal aggression among prisoners, which was reported by 18 participants. In the context of increasing disagreement of opinion, 9 participants reported that verbal aggression escalated to physical violence. A minority of participants (n = 3) reported seeing others involved in almost daily physical violence in any part of the prison (wing, yard, and cell). Sometimes violence involved only two protagonists, while other episodes involved several prisoners. Physical violence included boxing, stabbing, slashing, or 'blading' (with a razor). These events occasionally required hospital admission for one or more of those involved. In rare occurrences, some participants reported being aware of violent altercations that had resulted in the death of a prisoner. In addition, a small number of respondents expressed their concern regarding permanent physical harms, including loss of teeth and facial disfiguration. Sexual violence, such as rape (which may result in damage to the skin or mucosa and subsequent blood exposure), was not raised by participants when discussing violence.

> Yeah, I've seen all sorts. I've seen, you know, I've seen blokes get stabbed. I've seen all-in brawls. I've seen one-on-one fights. I've seen arguments. I've seen, you know, I've seen fights between officers and inmates. I've seen pretty much all forms of violence that there is.
>
> (Respondent 6, male, HCV negative, IDU and other risk exposed).

> I busted his mouth, smashed his tooth in, and it ripped up all my knuckles.
>
> (Respondent 16, male, HCV negative, other risk exposed).

Women participants reported either observing or hearing about particularly violent acts between prisoners.

> I've seen scissors go into a girl's temple. I've seen people get stabbed. I've seen hot water get thrown on somebody. I've seen a lot of bashings. I've seen broomsticks get thrown over somebody's head. Vacuum cleaners get thrown. I've seen chairs get thrown over somebody's head. Obviously just fists. A lot of fists. A screwdriver. Home-made shivs [a knife made in prison].
>
> (Respondent 8, female, HCV negative, IDU).

A few months ago here there was three ladies who went in. Girl got a drop and she–[She got drugs brought in]. They held her in the room, bashed her and then went up inside her to get the drop but there was no drop up there. She already had a mate holding onto it because there was out in the air that these girls were going around doing this to women in here. [Interviewer asked whether she was holding the drugs anally or vaginally]—Vagina.

(Respondent 24, female, incident HCV, IDU other risk exposed).

A majority of participants reported verbal disagreements quickly escalating into physical violence as a result of anger. With such altercations occurring sporadically and 'in the heat of the moment', people in prison do not have opportunity to contemplate whether their opponent has any blood borne viral infection or to consider the potential risk of acquiring infection. However, a few participants expressed self-restraint, holding back from fighting with other prisoners, because of their concern regarding HCV exposure.

Everyone makes an angry decision or snap [participant clicks fingers] on the spur of a second, but it takes insight to, you know, to, to sit there and go, "Fuck, if I [fight] this bloke, I know I'm gonna hurt this bloke. But, but if I split him [cause an open wound] then I split meself in the process and I get, does he have, does he have a blood-borne virus that I may then contract?" you know. It doesn't normally get to that point. People in here aren't, I can't say aren't capable of getting to that second level of thinking but it's, people let their anger control their, their level of thinking.

(Respondent 3: Male, HCV negative, IDU and other risk exposed).

Like I've been in situations before where, where I'm arguin' with a bloke that I know is a known scumbag, you know. And I mean he's mouthin' off and carryin' on. I'd love more than anything to just go and punch him in the head but, even to look at him, he looks like a disease-riddled scumbag, you know. So, you know, I'll hold back 'cause I don't, wouldn't want his blood to get–some scabby-lookin', fuckin' toothless fuckin' thing, you know. Like I don't want it, you know. I don't want his blood anywhere near me. So just try and shake it off or try and ignore it the best way you can.

(Respondent 5: Male, HCV negative, IDU and other risk exposed).

The concern regarding BBV transmission, including HCV, through physical violence was variable. Some participants were concerned about the potential transmission. One respondent described an incident in which he understood two people had contracted HCV following fighting. However, concerns of risk of HCV transmission through fighting were not consistent among all prisoners. Some prisoners had not considered, or were unaware, that HCV transmission can occur through fighting.

I've only known two guys [who] have contracted hep C through punch-ups. And, or through blood-to-blood contact through combat, yeah. So it's not that, you know, like needles take the cake when it comes to hep C spread. Like needles is the top, the top one but you do run the risk. If you bust your knuckles or your mouth gets busted by someone and blah, blah, blah, blood transference. If you get hit hard enough and both people are bleedin', it's gonna push the blood into your open wound. It's gonna get into your system. You're

gonna get infected. There's a high chance you won't but there's a high, you know, there's still the chance that you will.

(Respondent 18, Male, HCV negative, other risk exposed).

I thought about that before I come to jail. I had a friend that always worries about getting into a fight and catching hepatitis C. I don't know. I never really, it never really crossed my mind that you could catch it from fighting.

(Respondent 21, female, HCV negative, IDU other risk exposed).

Use of protective gear during fighting was inconsistent among participants. There was some mention of wearing protective gear during physical altercations. However, this required preparedness and planning. Planned physical altercations were much less frequent than spontaneous altercations.

The majority of the blokes I've seen fight, they've always got a couple of pairs of gloves on and, you know, they'll always take precaution. And 90 per cent of them don't have hepatitis you know. And they're good blokes but they're big men that will fight the right battle. Oh, if a fight goes down, you'll see, [. . .] by the blokes that have got gloves on, you know. They've got five pairs of latex gloves on, a pair of woolly gloves on. You know what I mean? Like why is he doing that in 40-degree heat? You know what I mean? Like you know he's, something's goin' on. So yeah, they always take precaution. And the officers come and check our knees and our hands and that, and see if we're split open.

(Respondent 11, Male, Incident HCV, IDU).

The decision about fighting with an individual was occasionally influenced by the HCV status of the opponent, with at least one participant indicating that a person with known HCV infection may be immune to violent altercations as potential opponents seek to protect themselves from possible exposure.

At the time, when you're punching on, I don't think girls even think twice about it. But there are a lot of fights that don't happen because of the fact that you know that girl's got hep C and you don't wanna be splitting your knuckle on their teeth or anything like that, or splitting them open and, you know, damaging your knuckles or hands in any way, or, to contract it. And so having hep C sometimes saves girls from getting a beating. I mean, but then again, I do know girls that have gone and put several pairs of gloves on before they've gone up and hit a certain person because they know she's got hep C. They usually just go and put the gloves on, and go and have a fight, yep.

(Respondent 20, Female, HCV negative, IDU only).

## 2. Network level

At the network level, the interpersonal relationships and social network among participants influenced some activities and perceptions that made them vulnerable to violence and subsequent HCV transmission. Participants' networks included those both within, and outside, prison. The major issues that led a participants to be vulnerable to violence in prison, included drug involvement, and reporting other prisoners' infractions to correctional officers, known associates or adversaries of incoming prisoners (including from community and prison

transfers). In addition to the physical injury to participants, these activities ultimately framed the social standing of that participant, making them vulnerable to be 'stood over' or subjected to intimidation including violence throughout their incarceration.

Drug use in the prison is costly and reliant on social networks [37]. As such, a strong coherent network is crucial to maintain drug and equipment supply in prison. However, drug dealing in prison makes people in prisons vulnerable to violence. A majority of the participants (n = 15) reported drug debt and the associated need for intimidation as a major cause of violence in prison.

> Drug debts is a big one, you know. Drug debts arise, people can't pay the debt. The guys know they're not gonna get the money—they smash him—but it's out of ego and pride. They have to do it 'cause, if they don't do it, they'll be perceived as weak. If they let him get away with it, even though he's a raging junkie, the guy's just, he's a fuckin' mess, it's like, "Oh well, he can't pay. He's gotta get smashed." And like guys might not go on with it so hard.
>
> (Respondent 18, male, HCV negative, other risk exposed).

Reporting others' wrongdoings in prison to correctional officers (such as drug dealing inside prison or an incidence of fighting among persons in prison) was regarded by participants as warranting physically violent punishment within the network. This behaviour was considered as a breach of trust to the other prisoners, which was therefore a violation of the largely unspoken inmate code of conduct [38].

> The number two [considering number one cause of violence in prison as drug debt] is somebody, you know, dobbin' on someone. Talkin' out of school or, or givin' up somebody else.
>
> (Respondent 6, male, HCV negative, IDU and other risk exposed).

The network level context of violence was also influenced by factors outside the prison, such as previous disputes and grievances between existing and incoming prisoners. These ongoing conflicts may escalate into physical violence.

> Now say if something happened on the outside that you got into an altercation with and you come in jail, now that person's in jail. You worry about, "Is he here? Am I gonna get into a confrontation? Are we gonna fight?" Do you know what I mean? That's one way of worrying about violence.
>
> (Respondent 1, male, HCV negative, IDU and other risk exposed).

## 3. Community level

The prison community was described as being stratified into groups based on racial identity and social status of the prisoners. People of specific community groupings, from same geographic region or cultural backgrounds were often bonded together inside prison. Within these groupings, people often followed a similar code of conduct. However, hierarchies and conflicts existed between groups, whereby the influence of one group or community over another made some groups more vulnerable to violence. Interestingly, some respondents indicated that the initial cause of violence may be issues like drug dealing, which then escalated to

become racial conflicts after involvement of people in prison of the same racial group, typically their peers, with whom they had already formed allegiances. However, in a few occasions, the racial influence was so strong that one respondent reported to be victimised for having previously participated in racially-motivated riots in the community [39].

> It was like an all-in brawl. Brothers [Aboriginal people] against Asians. Aboriginals against Asians. All over drugs that was. Like three of 'em got taken out of here on stretchers. Three or four of 'em. [. . .] There could have been about 13, 14 people involved. When [violence] breaks out, it breaks out.
>
> (Respondent 13, male, incident HCV positive, IDU).

> I nearly got killed over the [racially motivated riots in the community], so I nearly got stabbed in the throat—because they knew that I was a rioter.
>
> (Respondent 3, male, HCV negative, IDU and other risk exposed).

The groups inside prison were influenced by some motorcycle gang members, who were engaged in drug deal. The network level drug dealing was linked to the community level influence of the groups which predisposed physical violence in prison.

> Guys will come into the wing and it's like, "Oh yeah, well he's from a different bike group," or, "He's from a rival group around. . ." Like, cause there's a lot of gangs around south-west Sydney and, and around Sydney itself, and a lot of it's gang-related and drug-related. Most are bashings and stabbings again related, drug-related in gaol.
>
> (Respondent 18, Male, HCV negative, other risk exposed).

A social hierarchy exists within the prison based on the person's crime, whereby people who have committed specific crimes considered abhorrent, notably sexual assault of children, are vulnerable to violence from other prisoners.

> I've seen a few punches thrown but here it's lethal. They're, they're wild men. They'll jump on you, even 'til you're not movin'. They'll kick the shit out of ya. I seen it, you know. A week ago that happened to two blokes at once. And 10 blokes just got into 'em. And they didn't walk out. They found out they were in for child sex offences or something. Plus you get–heard that, you know, in the main, they'd have been hidin' there for 16 months and it's only just come out, but it come out. And yeah, they didn't want them in the yard so they got rid of them the only way they could. They could go and ask the officers to move 'em. The officers aren't gonna move 'em. So it comes to violence to get rid of 'em, and they leave their mark on them.
>
> (Respondent 11, male, incident HCV, IDU).

## 4. Policy and law level

Policies and laws relating to the prison setting are designed to control liberties, and can be enacted as a means to dissuade violence between people in prison and towards others. As punishment for perpetrators of repeated violence in prison, correctional officers often lock inmates in their cells for a period (lock-down) or place them in segregation (i.e. a separate cell with no opportunity to interact with other prisoners) and the security status (classification) made

more restrictive. Participants in segregation are also deprived of their standard prison privileges including phone communication with family and friends. These increased restrictions, resulting from violent altercations, can have significant social and emotional implications for people in prison.

> Oh if it kicks up again, there could be tension in the yard. Can be anything from gettin' locked in . . . You know what I mean? If it's a big enough altercation, they could, we could get locked in, locked down for a week. You know what I mean? You don't get phone calls— Oh well there goes your privileges like phone calls, anything. You know what I mean? Just in general.
>
> (Respondent 22, male, incident HCV, IDU other risk exposed).

Participants with an imminent parole hearing responded that they did not want to get into physical altercations, because such interactions might affect their chance of being released on parole (thereby reducing the time they are incarcerated in prison). A spurious allegation of fighting could also be used to defame a participant.

> [What are the things people worry about violence?] Gettin' tipped. Losin' their classification. Yeah, like gettin' sent to another gaol, you know. That's a big thing. The prison that you're at currently could be close to family, close for your girlfriend, you know. It could be good for visits. You get into a fight and then get sent somewhere like [a regional prison], then you're up shit creek. You get no visits then. So that could be a main thing. Like you could have parole coming up, you know. More charges laid on you could mean that you could have difficulties comin' up for parole, you know. But the main thing is, is people just don't want the drama. They don't wanna get hurt or whatever, or they don't want the, you know, the bullshit that comes afterwards.
>
> (Respondent 6, male, HCV negative, IDU and other risk exposed).

> Without the screws [correctional officers] finding out [about violence against other prisoners] because it will go against my name when I go up for parole in five months. You know, having a violent charge already it won't look good, you know.
>
> (Respondent 24, female, incident HCV, IDU other risk exposed).

The physical structures of prisons regularly include surveillance equipment such as cameras. This had implications for where and when violence occurred.

> You sort of get it [the violent incident] over and done with. The guys who are real, real proper, they won't sit there, "Well come in the cell, come in the cell." They don't care where the guy is. If he's sittin' in the middle of the wing in front of the cameras, whatever, he'll just run straight down. If he's got a problem with him then and there, he'll just go straight up and bang!
>
> (Respondent 18, Male, HCV negative, other risk exposed).

Prison policies mandate that disinfectant be used for cleaning up blood spills, such as spills following violence [40]. However, the supply and availability of proper cleaning equipment and disinfectant chemicals were not always optimal across all prisons.

One time we had the crystals [granules to aid in solidifying fluid for clean-up], another time we didn't—we just had to use bloody toilet paper to sop up most of the blood that was on the ground.

(Respondent 18, Male, HCV negative, other risk exposed).

Like this guy got stabbed once and then we were sweepers [prisoner cleaners] in the wing and the screws [officers] come in, and they said, "All right, here's the blood clean up kit." And you have to go and put little crystals [granules] on all the blood that's on the ground. And you've gotta glove up, sweep it up then Fincol [a disinfectant, bleach alternative] it out. See that's the other time is. . . inmates are expected to clean up the mess after.

(Respondent 18, Male, HCV negative, other risk exposed).

## Discussion

This qualitative study has identified contextual insights regarding violence at different levels of the social-ecological framework, describing perceptions of HCV transmission risks among those who are incarcerated. Our findings showed that physical violence in particular was inextricably intertwined with unique socio-ecological factors in the prison setting. The risk factors across the framework were complex, and inter-related with individual level risk factors impacting on, and impacted by, interpersonal network, community, and policy levels. Our study provides a unique integrated opportunity to frame the intricate context of HCV transmission in prison with violence as a key factor.

At the individual level of the framework, there was a variable degree of awareness and concern about the risks of HCV transmission associated with violent behaviours. There was considerable concern among some participants, evident in the practice of using protective gear where possible, such as in fights which were planned ahead. By contrast, participants also raised concerns about the impromptu nature of fights, which frequently occurred on the spur of the moment without the potential for personal protective equipment such as gloves. Our findings were consistent with another qualitative study in NSW correctional centres where people in prison were asked about concerns around reinfection of HCV whilst incarcerated [16]. The participants in that study perceived the risks of acquiring HCV infection through blood exposure during physical violence as being comparable in risk to injecting drug use. Previous studies have identified higher rates of victimization of violence of female prisoners than male prisoners [41,42]. Without our research, women reported greater episodes of violence than male participants. One instance described highlighted the vulnerabilities of women who smuggle in drugs via insertion. No similar occurrences were described among male participants. Several previous studies have explored the contexts and risk factors for engaging in violence in the prison setting irrespective of genders, which include younger age (≤21 years), being unmarried, prior incarceration, prior violent behaviours, use of drugs, and those who had depression or personality disorder, and gang involvement [43–48]. The context portrayed at the individual level of our framework represents similar socio-demography. Our findings suggest fighting is viewed as inevitable, and without means to adequately protect oneself against HCV transmission. Consequently, it is likely that HCV screening should be routinely offered to all people in prison who have engaged in physical violence in which blood exposure likely occurred, irrespective of other risk factors.

At the network level, drug debt was the major risk context that predisposed to violence. This is in-line with other studies which have found drug debts to be a major contributor to

violence in prisons [49–53]. Illicit drug use and drug dealing is inherently linked to criminali-sation activity that leads people to prison [54]. In this regard, coupled with risk factors for engaging in violence (e.g., incarceration, use of alcohol and/or drugs, history of violence), prison acts as a hotspot for violence and potential blood borne virus transmission [50]. Our data revealed specific instances of violence where the network of influence involved multiple prisoners and possibly also individuals external to the prisons. Other risk behaviours, such as disclosing information to correctional officers ('dobbing' on others), made people in prison vulnerable to violence due to violation of the unspoken prisoner code of behaviour [38]. These findings suggest that interventions against violence to reduce HCV transmission through net-works should be different than those targeting individual level factors. For example, legisla-tions regarding supply of drug and needle syringe exchange in prison setting should be intervened at policy level, rather than individual or network level.

At the community level, racial and other social influences on violence were evident, appar-ent across multiple ethnic groups such as Aboriginal or Arabic backgrounds, as well as motor-cycle gang members. Previous studies have identified comparable ethnic and socially driven violence amongst people in prison [55]. To prevent racial conflict, ethnic clustering (also known as 'yarding') has occasionally been implemented in NSW prisons. For example, at Goulburn Correctional Centre, yard 6 is allocated for Asian background prisoners, yard 7 for Islanders, and yard 8 for Arabic prisoners [56]. However, this ethnic clustering is done on *ad hoc* basis and not maintained in a well organised manner. Similar racial segregation has been applied in California prisons [57]. However, separating people in prison based on race is an ethical concern; a court order in California stipulated that racial segregation can only be imposed for an intermittent period in special circumstances, such as when an imminent racial conflict in prison seems likely [58]. Moreover, previous research which explored differences in social capital among Aboriginal and non-Aboriginal men in prison found bonding and linking social capital varied between the groups [37]; hence, bonding social capital, particularly among Aboriginal men in prison, could be utilised to promote appropriate health interventions. Col-lectively, these findings suggest that race can be a source of violence within prison, yet racial connections can also be a resource for health promotion. Prisons should consider the ways in which race manifests within individual prisons to ensure health benefits over harms to health and wellbeing (such as may be compromised by unnecessary segregation).

In addition, there was clear recognition that some crimes which had led to incarceration, notably sexual assault of children, were a provocateur for inciting violence. In addition, as per the inmate code of prison sub-culture, sex offenders are ranked at the bottom of the prisoner hierarchy, and were supposed to be beaten or killed; among those offenders, child molesters are ranked lowest [59,60]. These findings suggest that interventions against violence to reduce HCV transmission through community level should be different than those targeting individ-ual level factors. For example, people in prison convicted of sexual assault of children should be housed separately (network) versus enhanced provision of gloves and disinfectant to reduce blood exposure during fights. Although the correctional authorities in the NSW prisons occa-sionally practice separation of sex offenders, segregation of these prisoners upon prison entry might reduce violence within the prison.

At the policy legislative level of the framework, previous studies have explored the utility of rewards in the form of shortened sentences, for which people in prisons must meet some mini-mum standard of good behaviour [61,62]. Fulfilling this standard imposes an obligation on prisoners—such as abstaining from drug use or avoiding violence in prison [61]. By contrast, the prison regulations regarding people in prisons implicated in violent behaviours which impose additional punishments, such as segregation, deprivation of privileges, and deferred parole (and hence prolonged incarceration) may reduce the occurrence of violence among

some prisoners [63,64]. In addition, provision of personal protective gear, including disposable gloves for cleaning and supply of disinfectants for possible blood borne virus transmission prevention in different prisons are widely varied across the globe [65]. The variable nature of choosing protective gloves during physical violence in prison identified in our study suggests that adequate supply of protective gloves and disinfectants may reduce potential HCV transmission from bloody violence. However, as indicated by participants, the use of protective gear was not always an option as some fights were spontaneous, occurring on-the-spot without forward planning or opportunity for preparation against possible blood exposure. Policy-level decisions regarding supply of disinfectant may help the prevention of blood borne virus transmission in prisons. There should be constant supply of proper disinfectants and protective gear in prisons in a location accessible to prisoners.

This study had limitations. Although participants did not discuss the stage of the HCV epidemic in the prison setting, several HIV researchers have demonstrated how the epidemic stage of HIV is reflected through the individual, network and community level of the framework [21,26,66]. As expected, there was limited discussion by participants of the policy level implications on HCV transmission as this study only included prisoner participants and did not include other stakeholder participants such as correctional administrators and policymakers. Stakeholders, including prison healthcare providers and policymakers might consider appropriate strategy from the broader context of the intricate framework of violence in prison, e.g. prison health education and strengthening harm reduction programmes could be beneficial. At the time of data collection, women represented slightly less than 8% of the Australian prisoner population [36], though women participants comprised 35% of participants within our study.

The small qualitative study findings are not generalisable to other countries of the globe, especially considering the individual, network and community practice and concerns in multicultural Australian prisons. However, stratified purposeful sampling including prisoners who either had a history of ever injecting drug use or had non-injecting risk behaviours (including tattooing, piercing or fighting) and acute, chronic and negative HCV infection status of the prisoners might ensure better representation of the prisoners' perspective on violence and HCV transmission risk in the Australian prison setting.

Intervention strategies considering the complex risk contexts should be integrated at the policy level to improve correctional facilities' responses. Hepatitis C screening should routinely be offered to the people engaged in fighting. Like any part of the globe, illicit drug use and drug dealing, factors associated with violence, are unavoidable. The correctional authorities should ensure adequate supply and access of bleach or other disinfectant preparations to the people in prison, which might de-contaminate blood spills following violence, and injecting equipment during sharing. Enhanced communication and engagement between the prison population and correctional authority and timely segregation might reduce violence. Further studies might inform appropriate segregation. Programmes organised by correctional services and represented by the inmates, such as an inmate development committee [67] and violence prevention programmes should take initiative to prevent violence based on the outlined framework. Ultimately, the evidence-based interventions, particularly cost-effective public health interventions targeting violence at every level of MSEM should be aimed to reduce transmissions of HCV in the prison setting.

## Supporting information

**S1 COREQ checklist. Consolidated criteria for reporting qualitative studies (COREQ): 32-item checklist.**
(DOCX)

**S1 Appendix. Interview schedule—Violence part.**
(DOCX)

## Acknowledgments

The HITS-p investigators include: Andrew R Lloyd, Kate Dolan, Paul Haber, Carla Treloar, Fabio Luciani, Michael Levy, William Rawlinson, Greg Dore and Lisa Maher.

The authors thank the participants as well as Justice Health and Forensic Mental Health Network and Corrective Services NSW for their support and assistance in facilitating this project.

## Author Contributions

**Conceptualization:** Hossain M. S. Sazzad, Carla Treloar, Andrew R. Lloyd, Lise Lafferty.

**Data curation:** Hossain M. S. Sazzad, Carla Treloar.

**Formal analysis:** Hossain M. S. Sazzad.

**Funding acquisition:** Andrew R. Lloyd.

**Investigation:** Luke McCredie.

**Methodology:** Hossain M. S. Sazzad, Carla Treloar, Andrew R. Lloyd, Lise Lafferty.

**Project administration:** Andrew R. Lloyd.

**Resources:** Andrew R. Lloyd.

**Software:** Hossain M. S. Sazzad, Carla Treloar.

**Supervision:** Carla Treloar, Andrew R. Lloyd, Lise Lafferty.

**Validation:** Luke McCredie, Carla Treloar, Lise Lafferty.

**Visualization:** Luke McCredie.

**Writing – original draft:** Hossain M. S. Sazzad.

**Writing – review & editing:** Hossain M. S. Sazzad, Luke McCredie, Carla Treloar, Andrew R. Lloyd, Lise Lafferty.

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
