## [Decision Letter · Decision Letter 0]

11 Sep 2020

PONE-D-20-24308

Violence and hepatitis C transmission in prison – a modified social ecological model

PLOS ONE

Dear Dr. Sazzad,

Thank you for submitting your manuscript to PLOS ONE. After careful consideration, we feel that it has merit but does not fully meet PLOS ONE’s publication criteria as it currently stands. Therefore, we invite you to submit a revised version of the manuscript that addresses the points raised during the review process.

We look forward to receiving your revised manuscript.

Kind regards,

Ngai Sze Wong

Academic Editor

PLOS ONE

Journal Requirements:

Additional Editor Comments (if provided):

In 'Data availability', you have mentioned 'Yes - all data are fully available without restriction'. I am wondering how the data will be shared/availabile.

Reviewers' comments:

Reviewer's Responses to Questions

**Comments to the Author**

1. Is the manuscript technically sound, and do the data support the conclusions?

Reviewer #1: Yes

Reviewer #2: Yes

2. Has the statistical analysis been performed appropriately and rigorously? 

Reviewer #1: N/A

Reviewer #2: N/A

3. Have the authors made all data underlying the findings in their manuscript fully available?

Reviewer #1: Yes

Reviewer #2: No

4. Is the manuscript presented in an intelligible fashion and written in standard English?

Reviewer #1: Yes

Reviewer #2: Yes

5. Review Comments to the Author

Reviewer #1: Thank you for the opportunity to review this manuscript. This research article reports on HCV transmission risk and violence in the prison setting using the modified social-ecological model (MSEM). This topic is important to address. It generates useful knowledge and suggests practical interventions that could be applied in the prison setting.

Major considerations:

Results:

• The authors have defined the characteristics of each level of the MSEM mode. Currently, what is described for “people committed specific crimes” (line 295) is placed in the network level which does not quite fit in based on the definitions provided. Why not placing this group, who are socially stigmatised, in the community level?

• Line 335: The paragraph starting here needs review. The authors are trying to highlight the ways of reducing violence in prison but don’t describe the views / knowledge of participants regarding reduction of HCV transmission by reducing violence at the policy and law level.

Discussion:

• Please consider providing discussion of any differential pattern between male and female prisoners in terms of exposure to violence and HCV transmission risk, and the implications.

• As the data reported is based on a smaller proportion of females (8/23) versus males, it will be useful if the authors could address if the proportional differences are consistent with the setting observed

• Please also address the limitation of qualitative study in terms of generalizability

Minor considerations

Method:

• This should include interview venue / means to protect privacy during data collection.

• Please describe whether the data has been assessed if it has reached saturation.

• Please also describe in text of who has been involved in data analysis (whether a single or a team of researchers?)

• Line 174: Please consider removing the use of percentage which is less meaningful in qualitative study considering the small sample the study had.

Results: It will be useful to include the details of race when reporting the participants’ profiles.

Limitation: Please correct typo - “… HIV transmission form bloody violence….” to “… HIV transmission from bloody violence….” (line 462).

Reviewer #2: This is a well-written and organized manuscript presenting new and policy-relevant qualitative data about how prisoners understand and navigate the risks of contracting and living with HCV. Overall, the application of a modified social-ecological model (MSEM) to framing and analyzing the risks articulated by prisoners in qualitative interviews works as a solid frame for the piece. My critiques and suggestions for revisions center around (1) the way the methods were described and the need to engage more with why certain choices were made, (2) wanting a bit more information about the context of the larger study of which these qualitative interviews were a sub-study, (3) wanting to understand a bit more why the focus in this study was on violence (as opposed to other common alternatives like drug use and tattooing) as a means of HCV transmission, and (4) hoping the authors will engage with the actual or potential role of institutional actors (prison staff, bureaucrats) in transmission. I discuss each in turn in the attached review.

6. PLOS authors have the option to publish the peer review history of their article (what does this mean?). If published, this will include your full peer review and any attached files.

Reviewer #1: No

Reviewer #2: No

---

## [Author Response · Author response to Decision Letter 0]

23 Oct 2020

Violence and hepatitis C transmission in prison – a modified social ecological model

Response to academic editor’s comment:

Journal Requirements:

1. When submitting your revision, we need you to address these additional requirements. Please ensure that your manuscript meets PLOS ONE's style requirements, including those for file naming. The PLOS ONE style templates can be found at https://journals.plos.org/plosone/s/file?id=wjVg/PLOSOne_formatting_sample_main_body.pdf and https://journals.plos.org/plosone/s/file?id=ba62/PLOSOne_formatting_sample_title_authors_affiliations.pdf

Response: We have made the necessary changes to comply with PLOS ONE’s style requirements. 

2. We note that you have indicated that data from this study are available upon request. PLOS only allows data to be available upon request if there are legal or ethical restrictions on sharing data publicly. For information on unacceptable data access restrictions, please see http://journals.plos.org/plosone/s/data-availability#loc-unacceptable-data-access-restrictions

In 'Data availability', you have mentioned 'Yes - all data are fully available without restriction'. I am wondering how the data will be shared/availabile.

Response: Our apologies for the confusion. The response provided at submission was made in error. As is standard practice with publications reporting on qualitative data, the data will not be made available. 

Response to reviewer’s comments:

Reviewer #1: 

Thank you for the opportunity to review this manuscript. This research article reports on HCV transmission risk and violence in the prison setting using the modified social-ecological model (MSEM). This topic is important to address. It generates useful knowledge and suggests practical interventions that could be applied in the prison setting.

Major considerations:

Results:

1. The authors have defined the characteristics of each level of the MSEM mode. Currently, what is described for “people committed specific crimes” (line 295) is placed in the network level which does not quite fit in based on the definitions provided. Why not placing this group, who are socially stigmatised, in the community level?

Response: Thank you for this suggestion. We have relocated the “people committed specific crimes” violence piece from network level to the community level in both the abstract (page 2, line 28-30), result (from page 15-16, line 345-356 to page 17, line 384-395), and discussion (page 22-23, line 521-531) sections of the revised manuscript.

2. Line 335: The paragraph starting here needs review. The authors are trying to highlight the ways of reducing violence in prison but don’t describe the views / knowledge of participants regarding reduction of HCV transmission by reducing violence at the policy and law level.

Response: Thank you for this suggestion. We have revised the paragraph to better reflect participants’ experiences of the imposed restrictions which may be imposed following violent altercations. (Page 18, line 398-399).

Discussion:

3. Please consider providing discussion of any differential pattern between male and female prisoners in terms of exposure to violence and HCV transmission risk, and the implications.

Response: This is a great suggestion, thank you. We have added a quote to the Results “individual-level context” section (page 11, line 238-243). We have also added the following to the Discussion (page 20, line 467-471): Previous studies have identified higher rates of victimization of violence of female prisoners than male prisoners [1, 2]. Women reported greater episodes of violence than male participants. One instance described highlighted the vulnerabilities of women who smuggle in drugs via insertion. No similar occurrences were described among male participants. 

4. As the data reported is based on a smaller proportion of females (8/23) versus males, it will be useful if the authors could address if the proportional differences are consistent with the setting observed

Response: We have noted in the limitations section (page 24, line 557-559) that women are over-represented within our study compared with the prisoner population at the time of data collection. 

5. Please also address the limitation of qualitative study in terms of generalizability

Response: We have added the following to the Limitations section: The small qualitative study findings are not generalisable to other countries of the globe, especially considering the individual, network and community practice and concerns in multicultural Australian prisons. However, stratified purposeful sampling including prisoners who either had a history of ever injecting drug use or had non-injecting risk behaviours (including tattooing, piercing or fighting) and acute, chronic and negative HCV infection status of the prisoners might ensure better representation of prisoners’ perspective on violence and HCV transmission risk in the Australian prison setting. (Page 24, line 560-566) 

Minor considerations

Method:

6. This should include interview venue / means to protect privacy during data collection.

Response: We have added in the Methods: To protect participants’ privacy, interviews were conducted in a private clinic room in the absence of correctional officers. (Page 6, line 128-130)

7. Please describe whether the data has been assessed if it has reached saturation.

Response: Interview transcripts have been assessed for data saturation, with no new themes emerging in the final interviews. The responses regarding violence in prison from this subset of participants achieved saturation, hence the interview focus conducted among remaining prisoner participants shifted to HCV treatment. (Page 7, Line 152-155)

8. Please also describe in text of who has been involved in data analysis (whether a single or a team of researchers?)

Response: The following has been added to the Methods: CT and LM developed the first-round coding framework. The coding frame was then used to organise interview data within NVivo 12 (QSR International Pty Ltd. Version 12, 2018). HS and LL developed the secondary coding framework to elicit responses pertaining to the MSEM. LL conducted secondary coding of random samples to ensure consistency. (Page 7, Line 160-165).

9. Line 174: Please consider removing the use of percentage which is less meaningful in qualitative study considering the small sample the study had.

Response: We have removed the percentage. (Page 10, line 206)

10. Results: It will be useful to include the details of race when reporting the participants’ profiles. (Page 7, line 156, 157).

Response: We have added details of race of the participants in table 1. While in the USA there are a number of racial identities which are often included in demographic data, Australia often collects this information as a binary response: Indigenous or non-Indigenous. Thus, we have provided this information in-line with Australian standards. (Page 9)

11. Limitation: Please correct typo - “… HIV transmission form bloody violence….” to “… HIV transmission from bloody violence….” (line 462).

Response: We have corrected the typo. (Page 23, line 543)

Reviewer #2: 

This is a well-written and organized manuscript presenting new and policy-relevant qualitative data about how prisoners understand and navigate the risks of contracting and living with HCV. Overall, the application of a modified social-ecological model (MSEM) to framing and analyzing the risks articulated by prisoners in qualitative interviews works as a solid frame for the piece. My critiques and suggestions for revisions center around (1) the way the methods were described and the need to engage more with why certain choices were made, (2) wanting a bit more information about the context of the larger study of which these qualitative interviews were a sub-study, (3) wanting to understand a bit more why the focus in this study was on violence (as opposed to other common alternatives like drug use and tattooing) as a means of HCV transmission, and (4) hoping the authors will engage with the actual or potential role of institutional actors (prison staff, bureaucrats) in transmission. I’ll discuss each in turn below.

First, I had questions about methods throughout, starting with the answers to the standard PLOS questions about data. 

1. The authors note that data is available for “researchers who meet the criteria for access to confidential data.” What does this mean? It sounds like it means it is practically only available to the researchers. I certainly hope not just anyone can approach a human research ethics committee and meet requirements for access to confidential data. If this is qualitative data about a vulnerable population, there may well be reasons to prevent making it available, distinguishing it from other data sources that might be more readily shared. One alternative possibility could be providing a list of quotations used in analysis as an appendix. At any rate, the authors need to be clear about what is and is not available, why, and what the alternatives are. 

Response: We have updated this. Please see Response 2 above to the Academic Editor. 

2. Next, the authors note that a nurse assisting with the larger study about HCV of which this was a sub-study both recruited and interviewed participants. They also note written, informed consent was obtained, and participants received $10. Why was a nurse chosen for recruitment and interviews; were concerns about biased engagement and conflicts of interest raised and addressed, either in the process of making this methodological choice, or in the process of recruiting and conducting the interviews? (The conflict I imagine: a medical provider switching roles into researcher, obtaining much more personal information than usual or necessary from a patient, and being potentially biased by that information, especially if they learn a patient is violent or ignoring health recommendations; likewise, a patient might not be fully open with a medical provider about risky activities.) 

Response: In NSW, justice health comes under the auspices of the state health department (Justice Health & Forensic Mental Health Network, NSW Health) and is a separate entity from the correctional department (Corrective Services NSW, Department of Communities and Justice). As such, reporting mandates are different, allowing health practitioners to ask about risk factors without requiring reporting to the correctional authorities. Nurses working in the correctional setting undertaking qualitative research with people in prison have reported that being known to prisoner-participants in a health capacity prior to interview participation has aided in rapport building [3], enabling elicitation of richer data. In the instance of HITS-p, the study nurse was well trained and supervised by the HITS-p study investigators including CT who is an experienced qualitative social researcher. Moreover, the study nurse was also trained in psychology. Participants were asked to speak broadly of their personal experiences regarding violence, but also that encountered by their fellow inmates. This approach allowed participants to reflect on their own accounts, as well as what they have witnessed occur.

3. Also in terms of the structure of the study, why was written consent obtained, and will that written consent be maintained? At least in the United States, prison studies often obtain waivers of written consent to maximize anonymity and mitigate any risk of individuals being identified and persecuted for anything they might have said. 

Response: In Australian correctional centres, it is standard practice to obtain signed consent from prisoners participating in research whenever feasible, and was stipulated by the research ethics committee (the ‘IRB’) for this study. Although signed consent is obtained, full names are not stored with participant research data. As noted in the Methods, transcripts were checked for accuracy and de-identified, then audio recordings deleted with only de-identified transcripts retained. 

4. In terms of the participation incentive of AU$10 – how valuable is this to prisoner participants? In the United States, that could buy basic hygiene items, a few phone calls home, etc., and would be very meaningful – perhaps even too meaningful; I would like to see a discussion of how this amount was reached, and an analysis of how it struck the appropriate balance between crediting participants for their time without unduly incentivizing them to participate. 

Response: While remuneration for prisoner participation in behaviour and social research varies widely by state within the US [4], it is standard practice for prisoners in NSW correctional centres to receive remuneration for their participation in such studies as acknowledgement of their time and expertise. This amount was recommended by the research ethics committee as being sufficient to constitute ‘reimbursement for time and convenience’ (as was stated on the consent form), but insufficient to provide a strong or coercive incentive to participate. In practice, these monies are typically expended for ‘buy-ups’ – that is purchase of food or toiletry items not otherwise available in the prison. (Page 6, line 132-136)

5. How closed or open-ended was the interview guide the authors used? More detail about its structure (as opposed to just its substance) might be valuable, perhaps even it could be included in an appendix?

Response: Thank you for the suggestion. We have attached the interview guide as an appendix. (Page: 35-36) 

6. On page 6, line 123, in the methods section, the authors use the term PWID without spelling it out.

Response: We have spelled out PWID as people who inject drugs. (Page 3, Line 41) 

7. In their description of their analysis of the data, the authors say the transcripts were “read closely,” organized with a “coding frame,” and then re-coded based on the five layers of the MSEM. I have many questions. First of all, what happened between close reading and the coding frame? Were memos written? Were transcripts coded and re-coded, and was any interrater reliability established? How many codes were generated? Were they consolidated in some way? How did the codes relate to the MSEM? Was the MSEM something the authors had planned to apply from the beginning, structuring their interview questions and analysis, or was it something that emerged for the data? Was there any relationship between the coding frame and the MSEM, or were these separate projects, with other analyses of the interviews planned for other purposes?

Response: CT and LM developed the first-round coding framework. The coding frame was then used to organise interview data within NVivo 12 (QSR International Pty Ltd. Version 12, 2018). Memos were written between close reading of the transcripts and development of the MSEM coding framework. HS and LL developed the secondary coding framework to elicit responses pertaining to the MSEM. LL conducted secondary coding of random samples to ensure consistency and to establish interrater reliability. The primary or initial coding frame and the MSEM coding frame are separate projects with distinct analyses to interpret different aspects of HCV risks within the prison setting [5, 6]. However, the importance of violence was a primary interest of the project and is examined here in a secondary analysis using the MSEM. (Page 7, line 160-168)

8. Given the very small number of participants in this study, are the authors confident that the level of identifying detail they provide will not compromise anonymity? Providing a table of characteristics of participants (how many of each gender, age, other demographics collected, and how many fell into each category of risk behavior) could alleviate this concern, especially if there are a large enough number of people in each category (often 10, but, here 5 would seem reasonable).

Response: Yes, we are confident that the limited details we have included will not inadvertently identify any individuals. We thank the reviewer for the suggestion and have included a table (Table 1) with demographic and clinical characteristics of the participants. (Page 9) 

9. How many people were recruited for this study? How many refused?

Response: Among 30 participants of the broader qualitative study, the final seven participant interviews focused on decisions about HCV treatment and did not include discussion of violence as a risk factor for HCV transmission. Information is not available in relation to participants refusing invitation to participate. (Page 8, line 189-191)

10. Second, there was a major gap, to my mind, in the presentation of the background and methods, relating to the relationship between this sub-study, and the larger study of which it was a part. The authors describe the ten-year period of the larger study, but give little context about numbers recruited and/or the scale of the larger study, or even their own study. How many participants were recruited for the qualitative interviews? How many refused? Was 23 the target number, or were the authors hoping for more and faced some challenges in recruitment? How many prisoners participated in the larger, 10-year study? Was the qualitative study always envisioned as part of the larger study, or did it grow out of something learned over the course of the larger study? Generally, better situating this study would help to clarify methods, the relevance and representativeness of the findings, and also contextualize findings.

Response: We agree with reviewer’s comment. We have provided further detail on the 10 year long HITS-p cohort study and have included the following: A total of 590 people in prison were enroled in this cohort study [7]. Participants enrolled in the HITS-p study were eligible for the qualitative sub-study. The objective of the qualitative sub-study was to understand the broader contexts and concerns regarding HCV transmission in prison. Thirty participants were recruited in the sub-study. Information is not available in relation to participants refusing invitation to participate. (Page 8, line 191). Among the participants, a subset of participant interviews describing contexts and concerns regarding violence in prison and HCV transmission was analysed for this study. The remaining participant interviews focused on decisions about hepatitis C treatment. (Page 5, line 89 - 95).

The primary or initial coding frame and MSEM are separate projects with other analyses planned for other purposes. The broader study aimed to cover a number of aspects relating to hepatitis C risk in prison and other analyses were performed [5, 6]. However, the importance of violence was a primary interest of the project and is examined here in a secondary analysis using the MSEM. (Page 7, line 165-168) 

11. Third, the contextualization of this study in existing literatures was a bit light, making the focus seem more narrow than necessarily does the work justice. In some cases this “lightness” of literature came out in over-simplifications, e.g., “Due to the criminalization of drug use, people in prison have higher prevalence of HCV infection than those in the community” (page 3, lines 36-37). This is a strong claim simplifying multiple complex mechanisms of inequality and disparate treatment institutionalized in incarceration. First of all, are there not many reasons why people might come into prison with higher rates of HCV infection (all kinds of risk factors pre-prison beyond being criminalized drug users)? In fact, stating that drug use explains all HCV infection in a given population (like prisoners) runs the risk of implying that everyone with HCV is a drug user, a troubling implication. Secondly, are there not many reasons (as described in this article) why people in prison might subsequently incur higher rates of HCV infection, many more structural than individual, as this statement implies; e.g., abuse at the hands of other prisoners or staff, lack of adequate healthcare, lack of knowledge of risks, etc. In other cases, the “lightness” of the literature review came out in the focus of the piece on physical violence in prison as a means of HCV transmission. Can the authors elaborate, beyond the fact that prison violence has not been well-studied as a mechanism of HCV transmission, on why they focused on violence to the exclusion of other modes of transmission? Did they always intend to focus on violence and ask only about this in interviews? Was there some evidence from the larger study about the relevance of violence for transmission? Is there some sense that violence may be closely tied to other means of transmission? This is implied in the “2. Network level” findings section, but could be much further explored. For instance, are tattoos linked to violence (they certainly are associated by prison officials with gang membership in the United States, and prisoners often get them for protective reasons)? And/or is there some sense that public health interventions to mitigate HCV transmission in violent encounters might be beneficial in other ways in the prison environment?

Response: Thank you for the recommendations. We have rephrased the statement: Among people who inject drugs (PWID), estimated 52% have detectable antibodies against HCV, and 58% have a history of imprisonment across the world [8]. Multiple factors contribute to the higher prevalence of HCV infection in prison than those in the community with criminalisation of drug use being the major contributor. (Page 3, line 41 - 45) 

We have strengthened our literature review in the Introduction section and highlighted the justification for a better understanding of violence in prison setting (Page 3 line 54 - 59).

We clarified in the Methods section that, through the violence part of the interview guide (Appendix 1- page 35-36) we explored the type of violence, and situations that lead to violence, as well as prisoners’ concerns around violence (page 6, line 142 – page 7, line 144). We have added in the Introduction: The qualitative study in Australian prisons pertaining to economics of drug use and blood borne virus transmission examined physical violence, drug debt and potential HCV transmission as a complex interrelated issue. (Page 3, Lines 56-59)

Among our study participants (undertaken within Australian prisons), there were no instances depicted of tattoos being associated with violence. 

We agree with reviewer’s point that, public health interventions to mitigate HCV transmission in violent encounters might be beneficial in other ways in the prison environment. We have included the recommendation in the last sentence of the Discussion (Page 24, line 577-579).

12. Fourth, the underlying frame of this study seemed to be focused on prisoners as individuals, making somewhat free (if constrained) choices about exposing themselves to HCV and managing that exposure. This belies the reality of incarceration as an institution frequently so oppressive that free choices are more than constrained. Perhaps this is an American perspective, and Australian researchers might dis-abuse me of this and convince me Australian prisons are different, though I have read little to convince me of that. A few specific points along these lines: In American prisons, I have never heard of prisoners thinking about or fearing HCV transmission through violent interactions, but every correctional officer I know fears being infected with HCV by prisoners who spit at or otherwise attack them. Yet I know of now such cases of transmission in that manner. This disconnect seems important to address in this piece: how often is HCV transmitted among prisoners through violent interactions? Is this just a staff fear being operationalized in this study? Why were staff not included in this kind of analysis? Might they be included in future analyses? Relatedly, on pages 18-19, the authors discuss the possibility and pros and cons of ethnic clustering in prison, again writing as if prisoners are in control of ethnic tensions, when, in fact, much research (again in the United States, though I cannot imagine there are not comparable analyses of Aboriginal tensions in Australia) has documented how correctional staff leverage innate ethnic differences to encourage and amplify racial tensions, as a means of turning prisoners against each other rather than against their keepers. 

Response: Multiple risk factors contribute to person-to-person transmission of HCV. In the prison setting, these risk factors predominantly include injecting drug use, tattooing, piercing, barbering, and violence involving blood-to-blood contact. Due to the lack of focus on violence as a risk factor for HCV exposure and transmission in prison, studies to identify the attributable risk for violence in HCV transmission are rare.

Our study was limited to prisoner participants. Consequently, while staff would likely provide valuable insights, their involvement was beyond the scope of this research. We report only on the perspectives of people who incarcerated and cannot provide insight into the experiences or perceptions of those who work in the prison environment. We have highlighted this point in the limitations paragraph in Discussion (Page 23-24, line 552-557).

One of the co-authors is from the US. While we are aware of ethnic tensions being leveraged by correctional authorities in US prisons, the prison culture is different within Australia and such instances are infrequent. 

13. One additional minor note – page 14, lines 36-36: The discussion of lock-down/segregation felt out of place and disconnected from the rest of the analysis, though potentially fruitful to explore and better integrate. Though I have reviewed this piece with an eye for details and frameworks that need to be clarified, the study seems valuable and like an important contribution to the literature, and I would look forward to reading a revision.

Response: We have added the following: Further studies might inform appropriate segregation. (Page 24, line 574) 

Reference:

1. Wolff N, Shi J, Siegel JA. Patterns of victimization among male and female inmates: Evidence of an enduring legacy. Violence and victims. 2009;24(4):469-84.

2. McClellan DS, Farabee D, Crouch BM. Early victimization, drug use, and criminality: A comparison of male and female prisoners. Criminal justice and behavior. 1997;24(4):455-76.

3. Ferszt GG, Hickey J. Nurse researchers in corrections: A qualitative study. Journal of forensic nursing. 2013;9(4):200-6.

4. Smoyer AB, Blankenship KM, Belt B. Compensation for incarcerated research participants: diverse state policies suggest a new research agenda. American journal of public health. 2009;99(10):1746-52.

5. Treloar C, McCredie L, Lloyd AR, investigators HI-p. Acquiring hepatitis C in prison: the social organisation of injecting risk. Harm Reduct J. 2015;12:10. Epub 2015/04/24. doi: 10.1186/s12954-015-0045-2. PubMed PMID: 25903401; PubMed Central PMCID: PMCPMC4413553.

6. Treloar C, McCredie L, Lloyd AR. The Prison Economy of Needles and Syringes: What Opportunities Exist for Blood Borne Virus Risk Reduction When Prices Are so High? PLoS One. 2016;11(9):e0162399. Epub 2016/09/10. doi: 10.1371/journal.pone.0162399. PubMed PMID: 27611849; PubMed Central PMCID: PMCPMC5017673.

7. Cunningham E, Hajarizadeh B, Bretana N, Amin J, Betz‐Stablein B, Dore G, et al. Ongoing incident hepatitis C virus infection among people with a history of injecting drug use in an Australian prison setting, 2005‐2014: The HITS‐p study. Journal of viral hepatitis. 2017;24(9):733-41.

8. Degenhardt L, Peacock A, Colledge S, Leung J, Grebely J, Vickerman P, et al. Global prevalence of injecting drug use and sociodemographic characteristics and prevalence of HIV, HBV, and HCV in people who inject drugs: a multistage systematic review. The Lancet Global Health. 2017;5(12):e1192-e207.

---

## [Decision Letter · Decision Letter 1]

16 Nov 2020

Violence and hepatitis C transmission in prison – a modified social ecological model

PONE-D-20-24308R1

Dear Dr. Sazzad,

We’re pleased to inform you that your manuscript has been judged scientifically suitable for publication and will be formally accepted for publication once it meets all outstanding technical requirements.

Kind regards,

Ngai Sze Wong

Academic Editor

PLOS ONE

Additional Editor Comments (optional):

Reviewers' comments:

Reviewer's Responses to Questions

**Comments to the Author**

1. If the authors have adequately addressed your comments raised in a previous round of review and you feel that this manuscript is now acceptable for publication, you may indicate that here to bypass the “Comments to the Author” section, enter your conflict of interest statement in the “Confidential to Editor” section, and submit your "Accept" recommendation.

Reviewer #2: All comments have been addressed

2. Is the manuscript technically sound, and do the data support the conclusions?

Reviewer #2: Yes

3. Has the statistical analysis been performed appropriately and rigorously? 

Reviewer #2: N/A

4. Have the authors made all data underlying the findings in their manuscript fully available?

Reviewer #2: No

5. Is the manuscript presented in an intelligible fashion and written in standard English?

Reviewer #2: Yes

6. Review Comments to the Author

Reviewer #2: (No Response)

7. PLOS authors have the option to publish the peer review history of their article (what does this mean?). If published, this will include your full peer review and any attached files.

Reviewer #2: No

---

## [Editor Report · Acceptance letter]

19 Nov 2020

PONE-D-20-24308R1 

Violence and hepatitis C transmission in prison – a modified social ecological model 

Dear Dr. Sazzad:

I'm pleased to inform you that your manuscript has been deemed suitable for publication in PLOS ONE. Congratulations! Your manuscript is now with our production department. 

Kind regards, 

on behalf of

Dr. Ngai Sze Wong 

Academic Editor

PLOS ONE